# Determinants of Virological Failure in HIV Patients on Highly Active Antiretroviral Therapy (HAART): A Retrospective Cross-Sectional Study in the Upper East Region of Ghana

**Abdulai Abubakari** [1], **Habibu Issah** [1], **M. Awell Olives Mutaka** [1] and **Mubarick Nungbaso Asumah** [1,2,*]

1 Department of Global and International Health, School of Public Health, University for Development Studies, Tamale P.O. Box TL1350, Ghana
2 Ghana Health Service, Kintampo Municipal Hospital, Kintampo P.O. Box 192, Ghana
* Correspondence: nungbaso.asumah@uds.edu.gh; Tel.: +233-240-701-265

**Abstract:** Background: Even though highly active antiretroviral therapy (HAART) for HIV (Human Immune Deficiency) patients has considerably improved viral load suppression, more people still struggle to reduce viral loads. The aim of the study was to determine the associated factors of virological failure in HIV patients on antiretroviral therapy (ART) in the Upper East Region of Ghana. Methods: A retrospective cross-sectional study involving 366 participants aged 15 years and above who were on HAART for six (6) months or longer with viral load results in 2020. Bivariate and multiple logistic regression analyses were conducted to identify the determinants of virological failure among HIV patients at 95% confidence interval (C.I.) with a significant level pegged at a p value less than 0.05. Results: The prevalence of viral load failure was 47.0% and adherence to antiretroviral therapy was 62.6% among patients on HAART. The significant predicators of virological failure were basic education (AOR (adjusted odds ratio) = 7.36, 95% C.I = 4.91–59.71), High school/Vocational /Technical education (AOR = 4.70, 95% C.I. = 1.90–9.69), monthly salary/income < GHS 375.00 (AOR = 7.20, 95% C.I. = 1.73–29.95), duration on ART for <1 year (AOR = 0.27, 95% C.I. = 0.10–0.75), ART regimen (Tenofovir + Lamivudine + Efavirenz) (AOR = 3.26, 95% C.I. = 1.95–11.25), 3–5 times missed medication per month (AOR = 2.86, 95% C.I. = 1.34–6.08) and ≥6 missed medication per month (AOR = 23.87, 95% C.I. = 10.57–53.92). Conclusion: Educational status, salary/income, ART duration, ART combination regimen, and number of doses missed were statistically significantly associated with virological failure in patients on antiretroviral therapy. The majority of the respondents adhered to ART, which led to moderate viral load suppression but lower than the target for 2020. There is the need to strengthen the ongoing accelerated social behavior change communication among patients on ART to enhance adherence in order to attain the new UNAIDS target of 95% viral load suppression by 2030 in the Upper East Region of Ghana.

**Keywords:** determinants; HIV; highly active antiretroviral therapy; patients; virological failure

## 1. Introduction

Since the advent of highly active antiretroviral treatment (HAART) in 1996, the prognosis for patients with HIV ("human immunodeficiency virus") has significantly improved [1]. Despite this improvement, in 2019, nearly 38.0 million people had HIV worldwide in 2019, of which 25.7 million lived in Africa; the Sub-Saharan area of Africa now has the highest rate of HIV/AIDS ("human immunodeficiency virus/acquired immunodeficiency syndrome") infection worldwide. Thus, AIDS is a top global health problem especially in Africa [2]. The Joint United Nations Programme on HIV/AIDS (UNAIDS) together with the World Health Organization (WHO) encouraged nations and partners globally to transform the evidence-based benefits of antiretroviral therapy through a chain of ambitious strategic target-setting initiatives [3]. The major approach implemented by the United Nations (UN) to eliminate the HIV/AIDS epidemic was to improve patients' access to antiretroviral

therapy across the globe towards attaining the UNAIDS 95-95-95 (i.e., 95% of persons with HIV should be aware of their status, 95% of those diagnosed with HIV should be receiving continuous ART, and 95% of those taking ART should have suppressed viral load in 2030 [4].

Prolonged antiretroviral treatment (ART), which has improved viral suppression rates and continuously decreased HIV/AIDS-related mortality, has received considerable attention in the battle against HIV/AIDS [5]. Epidemiological studies have discovered that people receiving ART are linked with viral suppression [6]. However, several factors account for virological failure in people living with HIV on antiretroviral therapy in different regions. Earlier studies emphasized that factors that are linked to viral suppression in patients with HIV on antiretrovirals, include treatment adherence, ART regimen, gender, and duration on ART [7,8]. Other findings showed that virological failure was statistically significantly linked with educational status, being male, marital status, young age, multiple sexual partners, and unsatisfactory adherence [7,9–14]. A longitudinal study by Owusu et al. [15] in 2017 found that there was a 5.4% (1.4–20.9%) increased chance of virological failure among patients whose parents were not employed compared to patients whose parents were employed. In Ho Municipality in the Volta Region of Ghana, timely ART refill and patients on treatment for more than 3 years were significant with viral load suppression [16].

In 2016, the Ghana government adopted the Joint United Nations Programme on HIV/AIDS (UNAIDS) "treat all" policy and the 90-90-90 (i.e., 90% of persons with HIV should be aware of their status, 90% of those diagnosed with HIV should be receiving continuous ART, and 90% of those taking ART should suppress viral load) target to eradicate the HIV/AIDS epidemic by 2030 [4]. However, Ansah et al. [10] discovered 23.6% virological failure and a viral load suppression rate of 76.4% in HIV patients on ART in the Kumasi Metropolis. Though the result (76.4%) was less than the UNAIDS third 90% target for viral suppression of patients on ART, there was an increased proportion of virological failure (23.6%) compared to the 34% national achievement.

In 2019, the Ghana AIDS commission reported that a total of 342,307 were living with HIV/AIDS; out of this number, 64% were females. In 2019 alone, 20,068 new cases were recorded, with 79% being females [17]. The people living with HIV infection increased from 342,307 people in 2019 to 346,120 in 2021. In 2021, 18,928 new cases of HIV were recorded; this is less than the 20,068 recorded in 2019 [18]. According to the Ghana AIDS Commission (GAC) [19], Ghana achieved 58%-77%-68% (thus 58% of persons with HIV were aware of their status, 77% of those diagnosed with HIV were supplied continuous ART, and 68% of those taking ART suppressed viral loadof the 90-90-90 at the end of 2020. The Upper East Region attained the first two 90s but performed at about 51% on the third 90 at the end of 2020. According to the director of the Ghana AIDS commission (GAC) [19], though Ghana, as well as the Upper East Region, could not meet the 90-90-90 target in 2020, "the hope is that we know the barriers and we have the roadmap as a country to meet the target in the years ahead, maybe beyond 2020" to achieve the new UNAIDS 95-95-95 by 2030 [19].

The Upper East Region like many other regions in Ghana, has strived in the direction of achieving the UNAIDS 90-90-90 target by 2020 [10]; however, according to the literature by Jobanputra et al. [8], the proportion of HIV patients in the Upper East Region with viral load suppression, about 2168 (51%), by the end of 2020, fell short of the global target of a 90% suppression rate. This suggests that the viral load failure rate of about 49% was quite high among the limited population with HIV in the region. The Upper East Region faces increased HIV/AIDS transmission rates, including mother-to-child transmission, opportunistic infections, increased mortality, and decreased life span compared to other regions in Ghana [8]. The Upper East Region has a prevalence of 5776 cases, with new infections at 339, and a death rate of 230 per year [18]. Adult prevalence is 0.77%, with 4487 adults on antiretroviral treatment and a total coverage of 77.7% in 2020 [18]. The Upper East Region performed at about 51% on the third 90 at the end of 2020 [19].

According to Afrane et al. [20], limited evidence exists on the factors associated with viral load failure in patients on antiretroviral therapy. Currently, in developing countries including Ghana, inadequate evidence exists on the determinants of virological failure among HIV patients on HAART. Therefore, identifying factors limiting viral load suppression in HIV could lead to enhanced approaches that can lead to increased success of HIV treatment and improve the quality of life of HIV/AIDS patients in the Upper East Region and the world at large. It is against this backdrop that this study seeks to identify the determinants of virological failure in HIV patients on antiretroviral therapy (ART) in the Upper East Region of Ghana, as this could inform policy and decision making to attain 95-95-95 by 2030.

## 2. Methods and Materials

### 2.1. Description of Study Setting

The Upper East Region is located in the north eastern part of Ghana. It occupies a land mass of 8842 square kilometers. The region shares boundaries with Togo to the east and Burkina Faso to the north in West Africa. The Upper East Region also shares borders with the North East Region (previously part of the Northern Region) to the south and the Upper West Region in the west. At the end of 2020, the total population was 1,302,718, occupying 15 districts and municipalities. The total number of health facilities in the Upper East Region of Ghana is 547 (consisting of 20 hospitals [1 regional hospital at Bolgatanga, 6 district and municipal government hospitals, 10 private hospitals, and 3 Christian Health Association of Ghana (CHAG) hospitals], 67 health centers, 38 clinics, 419 Community Health Planning and Services (CHPS) compounds, and 3 private maternity homes). Out of the 547 health service centers, 28 provide antiretroviral services, with Bolgatanga Regional Hospital as the only center with the capacity to test viral load.

### 2.2. Study Design

The study employed a retrospective cross-sectional study design because it allowed us to obtain a "snapshot" of the factors influencing virological failure in HIV patients receiving highly active antiretroviral treatment in the Upper East Region.

### 2.3. Inclusion and Exclusion Criteria

The region had 4315 active patients on antiretroviral treatment for 6 months and longer at the end of 2020. Patients who were on antiretroviral therapy for ≥6 months with at least one (1) readily available viral load result had the chance to be included in the study. However, HIV patients lost to follow up were not included.

### 2.4. Sample Size Determination

The study's sample size was estimated using Survey Monkey [21]. The targeted population (N) of 4315 active patients on antiretroviral treatment for 6 months and longer at the end of 2020 was used. Using a confidence level of 95%, 5% margin of error, and 50% response distribution, the sample size was estimated to be 353. To cater for incomplete questionnaires, the sample size was increased to 366. To recruit the sample, the study used the multistage sampling technique. First, simple random sampling with a lottery selection technique was used to choose eight (8) ART clinics from the twenty-eight (28) ART service centers for the study. The same method was used to select four (4) facilities from the remaining 20 ART for pretesting and piloting of the data collection tool. For each of the selected facilities, the consecutive sampling technique was applied to sample 366 participants. Therefore, on a typical data collection day in a given facility, all those who came to access care and met the selection criteria and consented to be part of the study were interviewed. This process continued until the sample size was reached. The number of participants recruited in each facility was directly proportional to the total number of ART active clients with viral load tests. Table 1 shows the sample size allocated according to the total number of ART active clients with viral load tests.

**Table 1.** Sample size allocation according to the population of facilities.

| Name of Facility | Total ART Active Clients with Viral Load Test | Sample Size |
|---|---|---|
| Regional Hospital, Bolgatanga | 1274 | 108 |
| War Memorial Hospital | 759 | 64 |
| Bongo Hospital | 737 | 63 |
| Zebilla Hospital | 426 | 36 |
| Sandema Hospital | 379 | 32 |
| Tongo Hospital | 268 | 23 |
| Kongo-Logre Health Centre | 267 | 23 |
| Paga Hospital | 205 | 17 |
| Total | 4315 | 366 |

*2.5. Data Collection Tools, Procedures, and Data Management*

Quantitative data on HIV patients on antiretroviral therapy were obtained from the facilities' ART clinics. Viral load data was retrieved from the patient's folder, viral load registers, or the E-Tracker database. Socio-demographic characteristics, patients' behavioral information, age at the time of diagnoses, duration on ART, type of ART regimen administered, pills missed, and viral load tests done were all relevant for data collection. Structured questionnaires were administered to the respondents using Kobo Toolbox mobile app in English. Respondents who could read and write were given the questionnaire to fill, while those who could not read and write were assisted by translating the questionnaire to Frafra, Kassena, Nabd, Dagbani, and Hausa, the local languages commonly spoken in the Upper East Region.

Prior to the data collection, permission was sought from the Regional Health Directorate. In the recruitment period, participants completed the structured survey questionnaire upon consenting to the study. The survey questionnaires were administered by ART data managers and clinicians who were trained in data collection and who understood the local dialects, including Frafra, Kassena, Nabd, Dagbani, and Hausa, in the Upper East Region. One clinical staff member from each of the eight (8) targeted ART facilities was recruited and trained to collect the data. The data enumerators were trained to administer the questionnaire without any interference. The clinicians and ART data managers were selected because they had been working with these patients. Thus, since there are stigma issues regarding HIV/AIDS, it was assumed that the study participants would be more comfortable with health care providers. The principal researcher audited and verified the collected data for completeness, quality, and verification. A pre-test was carried out prior to the actual data collection to measure the validity and consistency of responses in the questionnaire on eight (8) HIV patients in four different health centers who met the inclusion criteria but were not part of the study population. All issues realized on the study tool and processes during the pretest were addressed before the study started.

To ensure quality data management, the researcher employed some mechanisms to reduce data bias. The research assistants (the selected clinicians and ART data managers) were trained on Kobo-Collect Software, data collection procedures, and the content and subject matter of the questionnaire. Ten days before the start of the study, the questionnaire was pretested to enable the data collectors to become familiar with the tools. Data collection was done by eight (8) skilled ART staff and supervised by the principal investigator. Daily supervision was done on site, and questionnaires were reviewed and crosschecked by the investigator for consistency, completeness, and accuracy and discussed with the data collectors. The researcher carried out data cleaning using Microsoft Excel and proceeded to data analysis.

*2.6. Study Variables*

The main outcome (dependent) variable in the study was viral non-suppression status in HIV clients on antiretroviral therapy. In the Low- and Middle-Income countries

(LMICs), the WHO defined viral load suppression to be <1000 copies/mL and viral non-suppression to be >1000 copies/mL, while a viral outcome of <50 copies/mL was defined as undetectable [22].

The independent variables were classified into social-demographic characteristics (gender, age, level of education, employment, marital status, and salary/income), behavioral factors (alcohol consumption, number of sexual partners, and condom use), and other variables such as adherence to counseling, ART duration, ART regimen combination, challenges in receiving timely ART, number of doses missed, and ART side effects. Other variables included spouse/partner HIV status and number of viral load tests completed.

### 2.7. Data Analysis

Data analysis was conducted using Statistical Package for Social Sciences (IBM SPSS Inc, Chicago, IL, USA) version 26.0. Descriptive statistics such as frequencies and percentages were generated and presented in tables. Using the WHO ART adherence guideline, within a month, patients who missed ≤2 doses of antiretroviral therapy were said to have good adherence level (≥95%), patients who missed 3–5 doses had a fair adherence level (≥85% to <95%), and patients with ≥6 doses missed were said to have a poor adherence level (<85%) [23]. Other studies have revealed that ≥95% is the required drug adherence level for successful treatment outcomes [24].

HIV viral load achievement was categorized into three (3) types, including target not detected or not detected (<50 copies/mL), virally suppressed (<1000 copies/mL), and viral failure (≥1000 copies/mL) [10,19,22]. Consistent with the available literature [22,25,26], viral load achievement was re-grouped into two (2); viral load suppression (<1000 copies/mL) and virological failure (≥1000 copies/mL). This study analyzed and measured virological failure against other variables, including treatment adherence, ART duration, ART regimen type, education, etc.

A chi-square ($X^2$) test was used to determine the association between the various variables and virological failure among patients on ART. Using multiple logistic regression analysis, the study determined the relation between the dependent variables and the independent variables. To achieved a goodness of fit for the model, only variables with a *p*-value of <0.25 were included in the multiple logistic regression model. A confidence level of 95% during the study was administered with a 5% level of significance.

### 2.8. Ethical Considerations

Permission for ethical clearance was sought from the Ghana Health Service Ethics Review Committee (GHS-ERC:047/03/22) to carry out the research. The Regional Health Directorate of the Ghana Health Service (GHS) also granted the researcher permission to carry out the study in the Upper East Region. The purpose of the study was described to all the respondents in order to allow an informed decision to participate or not to participate in the study. All participants gave an oral consent and signed a written informed consent form after obtaining all study information. However, parental or guardian approval was required for individuals under the age of 18. In this case, the consent form was duly signed. Additionally, because of the nature of the study, participants were assured of the efforts to safeguard their privacy, and confidentiality was ensured. Participants in the research who refused to give their permission were not enrolled. The study's participants were made aware that taking part was completely voluntary and that they may withdraw at any time if they so wished. Participants received no monetary compensation for participating.

## 3. Results

### 3.1. Socio-Demographic Characteristics of Respondents

The study recruited 366 respondents, with a higher proportion (37.2%) being within the ages of 25 to 34 years. The majority of the respondents (73.2%) were females, 56.8% were married, 52.5% had never been to school, 62.6% were self-employed, and 60.7% earned a

salary/income lower than GHS 375.00 a month. Table 2 shows the socio-demographics characteristics of the respondents.

**Table 2.** Socio-demographic characteristics of the respondents (n = 366).

| Variables | Frequency (n) | Percentage (%) |
|---|---|---|
| **Age group (years)** | | |
| 15–24 | 46 | 12.6 |
| 25–34 | 136 | 37.2 |
| 35–44 | 107 | 29.2 |
| 45–54 | 55 | 15.0 |
| ≥55 | 22 | 6.0 |
| **Gender** | | |
| Male | 98 | 26.8 |
| Female | 268 | 73.2 |
| **Marital Status** | | |
| Married | 208 | 56.8 |
| Never married | 62 | 16.9 |
| Divorced/Separated/Widowed | 96 | 26.2 |
| **Educational Status** | | |
| No formal education | 192 | 52.5 |
| Basic | 78 | 21.3 |
| SHS/Voc/Tech | 70 | 19.1 |
| Tertiary | 26 | 7.1 |
| **Occupation** | | |
| Public/Private employment | 46 | 12.6 |
| Self-employed | 229 | 62.6 |
| Unemployed | 91 | 24.9 |
| **Salary/Income Level** | | |
| <GHS 375.00 | 222 | 60.7 |
| GHS 375–1000 | 99 | 27.0 |
| >GHS 1000 | 45 | 12.3 |
| **Alcohol Intake** | | |
| Yes | 99 | 27.0 |
| No | 267 | 73.0 |

SHS/Voc/Tech: Senior High School/Vocational/Technical; 100USD ≈ GHS 1028.

### 3.2. Antiretroviral Therapy Adherence

The study showed that 49.2% of the respondents were diagnosed HIV within the ages of 25 to 34 years. A total of 97.3% of respondents received adherence counseling before starting their treatment. The study also revealed that 58.7% of the participants were on HIV treatment for 3 years or more. The majority of the respondents (76.5%) were on a Tenofovir + Lamivudine + Dolutegravir (TLD: 300 mg + 300 mg + 50 mg) regimen. In addition, 89.1% of patients found it easy to take the antiretrovirals (ARVs) without complaint, while 84.2% of the respondents said they received the antiretrovirals in a timely manner. The majority of the respondents (89.6%) did not experience any side effects from the antiretrovirals (Table 3).

A greater proportion of respondents (62.6%) missed ≤ 2 doses (i.e., ≥95% adherence) of antiretrovirals within a month, 12% missed doses within the range of 3–5 (i.e., <95%–≥85.0%) adherence, while 25.4% missed 6 or more (i.e., <85.0% adherence) doses of antiretrovirals within a month. Figure 1 shows the ART adherence among respondents.

**Table 3.** Antiretroviral Therapy Adherence (n = 366).

| Variables | Frequency (n) | Percentage (%) |
|---|---|---|
| **Age at HIV Diagnosis** | | |
| 15–24 | 155 | 42.3 |
| 25–34 | 180 | 49.2 |
| 35–44 | 31 | 8.5 |
| **Adherence Counseling** | | |
| Yes | 356 | 97.3 |
| No | 10 | 2.7 |
| **ART Duration** | | |
| <1 year | 36 | 9.8 |
| 1–3 years | 115 | 31.4 |
| ≥3 years | 215 | 58.7 |
| **ARV Combination** | | |
| Tenofovir + Lamivudine + Efavirenz | 65 | 17.8 |
| Tenofovir + Lamivudine + Dolutegravir | 280 | 76.5 |
| Zidovudine + Lamivudine + Dolutegravir | 21 | 5.7 |
| **Ease of taking ART** | | |
| Easy | 326 | 89.1 |
| Difficult | 40 | 10.9 |
| **Issues with Receiving ART in Timely Manner** | | |
| Yes | 58 | 15.8 |
| No | 308 | 84.2 |
| **Reasons Doses Missed** | | |
| Stigma | 22 | 6.0 |
| Felt sick/ill | 44 | 12.0 |
| Side effects | 38 | 10.4 |
| Forgot to take | 67 | 18.3 |
| Ran out of pills | 181 | 49.5 |
| **ART Side Effects** | | |
| Yes | 38 | 10.4 |
| No | 328 | 89.6 |

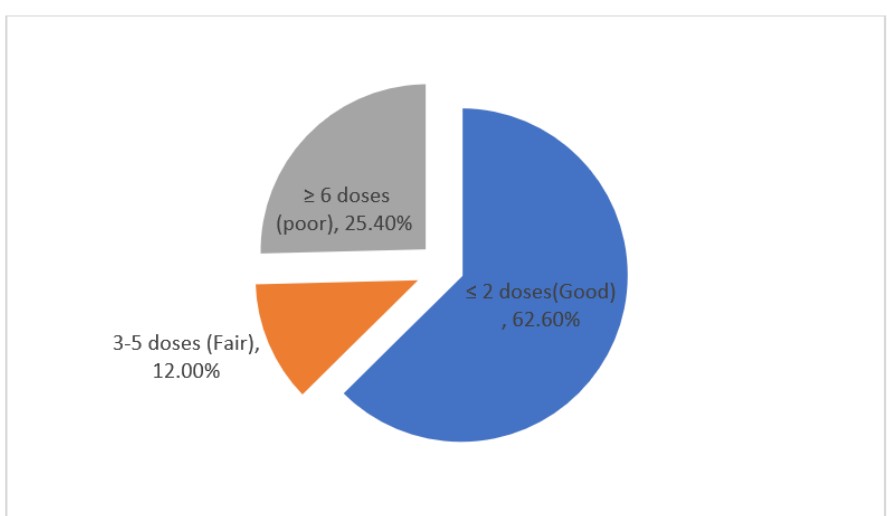

**Figure 1.** Number of doses missed by patients on antiretroviral therapy.

*3.3. HIV Viral Load Testing of Respondents*

The majority of the respondents (97.3%) received enhanced adherence counselling, while 69.1% completed only one (1) viral load test. Slightly above half of the respondents (53.0%) achieved viral suppression (<1000 copies/mL and Target Not Detected). Viral load results were interpreted for 92.6% of the respondents. The respondents indicated the following implications of a failed viral load (≥1000 copies/mL): opportunistic infection

(35.5%), caused new HIV infections (35.2%), reduction in survival rate (33.1%), HIV could lead to AIDS (31.7%), and could cause drug resistance (5.5%) (Table 4).

**Table 4.** HIV viral load testing of respondents (n = 366).

| Variables | Frequency (n) | Percentage (%) |
|---|---|---|
| **Received Enhanced Adherence Counseling** | | |
| Yes | 356 | 97.3 |
| No | 10 | 2.7 |
| **Number of Viral Load (VL) Tests Done** | | |
| 1 time | 253 | 69.1 |
| 2 times | 105 | 28.7 |
| 3 times | 8 | 2.2 |
| **Reasons for checking VL test *** | | |
| Check disease progression | 144 | 39.3 |
| Monitor response to ART | 333 | 91.0 |
| Check infectious level | 48 | 13.1 |
| Viral Load Results | | |
| <1000 copies/mL | 131 | 35.8 |
| ≥1000 copies/mL | 172 | 47.0 |
| Target Not Detected | 63 | 17.2 |
| **Results Interpreted to Respondent** | | |
| Yes | 339 | 92.6 |
| No | 27 | 7.4 |
| **Viral Load Result Status** | | |
| VL Suppressed | 194 | 53.0 |
| VL Failed (Unsuppressed) | 172 | 47.0 |
| **Implication of VL Failure *** | | |
| Can cause further infection | 129 | 35.2 |
| Increase to stage 3/AIDS | 116 | 31.7 |
| Opportunistic infection | 130 | 35.5 |
| Drug resistance | 20 | 5.5 |
| Reduce survival rate | 121 | 33.1 |

* Multiple response questions.

### 3.4. Determinants of Viral Load Failure in HIV Patients on Antiretroviral Therapy (ART)

In the adjusted model, respondents with basic education (AOR: 7.36, 95% CI: 4.91–59.71), and senior high school (SHS)/vocational/technical education (AOR: 4.70, 95% CI: 1.90–9.69) were 7.3 and 4.7 times more likely to have virological failure, respectively, compared to their counterparts with tertiary education.

Respondents who earned less than GHS 375.00 were 7.2 times more likely to have virological failure (unsuppressed) compared to those earning more than GHS 1000.00 (AOR: 7.2, 95% CI: 1.73–29.95).

Those who started the ART therapy less than a year ago were 73.0% less likely to have virological failure compared with those who had been on treatment for more than 3 years (AOR: 0.27, 95% CI: 0.10–0.75).

Respondents who were on Tenofovir + Lamivudine + Efavirenz combined HIV therapy were about 3.3 times more likely to have virological failure compared to those using Zidovudine + Lamivudine + Dolutegravir combined therapy (AOR: 3.26, 95% CI: 1.95–11.25).

Finally, the study revealed that respondents who had fair (i.e., those who missed 3 to 5 doses of ART regimen in a month) (AOR: 2.86, 95%CI: 1.34–6.08) and poor (i.e., missed 6 doses or more of ART regimen in a month) (AOR: 23.87, 95% CI: 10.57–53.92) were 2.86 and 23.87 times more likely to have a failed viral load test compared to those with good adherence to ART regimen (i.e., missed 2 or less doses of ART regimen in a month) (Table 5).

**Table 5.** Determinants of viral load failure in HIV patients on antiretroviral therapy (ART).

| Variables | Categories | Viral Load (VL) Status | | *p*-Value | COR (95% CI) | AOR (95% CI) |
|---|---|---|---|---|---|---|
| | | VL Suppressed | VL Failed (Unsuppressed) | | | |
| **Gender** | | | | 0.232 | | |
| | Male | 57 (58.2%) | 41 (41.8%) | | Ref * | Ref * |
| | Female | 137 (51.1%) | 131 (48.9%) | | 1.33 (0.83–2.12) | 1.04 (0.56–1.95) |
| **Age** | | | | 0.016 | | |
| | 15–24 | 28 (60.9%) | 18 (39.1%) | | Ref * | Ref * |
| | 25–34 | 59 (43.3%) | 77 (56.6%) | | 2.03 (1.02–4.01) * | 1.79 (0.62–5.16) |
| | 35–44 | 66 (61.7%) | 41 (38.2%) | | 0.97 (0.48–1.96) | 1.28 (0.44–3.79) |
| | 45–54 | 26 (47.3%) | 29 (52.7%) | | 1.74 (0.78–3.84) | 1.81 (0.54–6.08) |
| | ≥55 | 15 (68.2%) | 7 (31.8%) | | 0.73 (0.25–2.13) | 1.01 (0.22–4.71) |
| **Marital status** | | | | 0.374 | | omitted |
| | Married | 115 (55.3%) | 93 (44.7%) | | Ref * | omitted |
| | Never married | 34 (58.4%) | 28 (45.2%) | | 1.02 (0.58–1.80) | omitted |
| | Divorced/Separated /Widowed | 45 (46.9%) | 51 (53.1%) | | 1.40 (0.86–2.28) | omitted |
| **Education** | | | | *p* < 0.001 | | |
| | No formal education | 89 (46.4%) | 103 (53.6%) | | 8.8 (2.57–30.54) ** | 4.30 (0.94–34.34) |
| | Basic | 40 (51.3%) | 38 (48.7%) | | 7.28 (2.02–26.25) ** | 7.36 (4.91–59.71) * |
| | SHS/Vocational /Technical | 42 (60.0%) | 28 (40.0%) | | 5.11 (1.40–18.65) * | 4.70 (1.90–9.69) * |
| | Tertiary | 23 (88.5%) | 3 (11.5%) | | Ref * | Ref * |
| **Occupation** | | | | 0.019 | | |
| | Public/Private employment | 33 (71.7%) | 13 (28.3%) | | 0.35 (0.16–0.76) ** | 2.05 (0.52–8.09) |
| | Self-employed | 118 (51.5%) | 111 (48.5%) | | 0.84 (0.52–1.37) | 1.28 (0.63–2.64) |
| | Unemployed | 43 (47.3%) | 48 (52.7%) | | Ref * | Ref * |
| **Monthly salary/income** | | | | *p* < 0.001 | | |
| | <GHS 375.00 | 95 (42.8%) | 127 (57.2%) | | 6.18 (2.75–13.88) *** | 7.20 (1.73–29.95) ** |
| | GHS 375–1000 | 62 (62.6%) | 37 (37.4%) | | 2.76 (1.16–6.56) ** | 2.02 ((0.51–7.91) |
| | >GHS 1000 | 37 (82.2%) | 8 (17.8%) | | Ref * | Ref * |
| **Alcohol intake** | | | | 0.719 | | Omitted |
| | Yes | 54 (54.5%) | 45 (45.5%) | | 1.09 (0.69–1.73) | Omitted |
| | No | 140 (52.4%) | 127 (47.6%) | | Ref * | Omitted |
| **Age at HIV diagnosis (years)** | | | | 0.071 | | |
| | 15–24 | 87 (56.1%) | 68 (43.9%) | | 1.64 (0.72–3.72) | 1.74 (0.51–5.86) |
| | 25–34 | 86 (47.6%) | 94 (52.2%) | | 2.30 (1.02–5.15) * | 2.42 (0.77–7.61) |
| | 35–44 | 21 (67.7%) | 10 (32.3%) | | Ref * | Ref * |
| **ART duration** | | | | 0.017 | | |
| | <1 year | 27 (75.0%) | 9 (25.0%) | | 0.32 (0.15–0.72) ** | 0.27 (0.10–0.75) * |
| | 1–3 years | 61 (53.0%) | 54 (47.0%) | | 0.86 (0.55–1.35) | 0.70 (0.39–1.27) |
| | >3 years | 106 (49.3%) | 109 (50.7%) | | Ref * | Ref * |
| **ART combination** | | | | *p* < 0.001 | | |
| | Tenofovir + Lamivudine + Efavirenz | 20 (30.8%) | 45 (69.2%) | | 3.66 (1.31–1.21) * | 3.26 (1.95–11.25) ** |
| | Tenofovir + Lamivudine + Dolutegravir | 161 (57.5%) | 119 (42.5%) | | 1.20 (0.48–2.99) | 1.20 (0.39–3.72) |
| | Zidovudine + Lamivudine + Dolutegravir | 13 (61.9%) | 8 (38.1%) | | Ref * | Ref * |

**Table 5.** *Cont.*

| Variables | Categories | Viral Load (VL) Status | | *p*-Value | COR (95% CI) | AOR (95% CI) |
|---|---|---|---|---|---|---|
| | | VL Suppressed | VL Failed (Unsuppressed) | | | |
| **Missed doses/ ART adherence** | | | | *p* < 0.001 | | |
| | <2 doses (good) | 161 (70.3%) | 68 (29.7%) | | Ref * | Ref * |
| | 3–5 doses (fair) | 22 (50.0%) | 22 (50.0%) | | 2.37 (1.23–4.56) ** | 2.86 (1.34–6.08) ** |
| | ≥6 doses (poor) | 11 (11.8%) | 82 (88.2%) | | 17.65 (8.85–35.20) *** | 23.87 (10.57–53.92) *** |

Ref *: Reference; *p* < 0.05 *; *p* < 0.01 **; *p* < 0.001 ***; COR: Crude Odds Ratio; AOR: Adjusted Odds Ratio; CI: Confidence Interval.

## 4. Discussion

This study aims to determine the factors associated with virological failure in HIV patients on highly active antiretroviral therapy in the Upper East Region. This study found that educational status, salary/income, ART duration, ART combination regimen, and number of doses missed were statistically significantly associated with virological failure. The majority of the respondents adhered to ART regimen, and the viral suppression rate was slightly more than half.

The study showed that patients on antiretroviral therapy with basic and SHS/Vocational /Technical education were more likely to have failed viral suppression compared to patients who attained tertiary education. This result is consistent with other studies that showed that patients on antiretroviral therapy with lower or no education experienced high viral load outcomes [27]. Ansah et al. [10] in Kumasi also observed that lower levels of education increased the risks of virological failure in HIV patients on antiretroviral therapy [10]. This could be due to the fact that people with higher education experience better health, including self-reported health, healthy lifestyle, and low mortality, morbidity, and disability [28]. However, some other studies have highlighted that treatment success (viral load suppression) is continuously associated with the education of clients on ART [29]. Thus, the situation can be reversed with improved adherence counseling, health education, demand creation for viral load testing, and relevant information on antiretroviral therapy for patients with lower or no education.

The study showed that lower income was a key determinant of viral failure, which could be explained by the inability to meet basic essentials like healthy food and shelter, resulting in missed doses and greater risk of viral load failure [30]. This finding is consistent with the study in Harare, Zimbabwe, which posited that low income was associated with high viral load (≥1000 copies/mL) in patients on antiretroviral therapy [9]. We are of the view that to increase viral suppression, HIV patients with low income should be supported financially to motivate them to adhere to the treatment protocol. Some of these patients have to travel far distances due to stigma to access their medication.

Statistically, the study found a significant relationship between the duration (<1-year) on antiretroviral therapy and viral load detection compared to patients on ART for more than 3 years. Several studies have found similar results. For example, in South Africa, a study reported that patients on antiretroviral therapy from 6–12 months had increased odds of virological failure compared to patients in treatment for longer periods [31]. Additionally, a multi-country study in Vietnam showed an 88.5% viral suppression rate among clients on ART after 1 year compared to those in treatment for less than 1 year [32]. Possible reasons might be that patients may forget to take their medication at the initial stages or resort to traditional and spiritual healing treatment. As suggested by some researchers, the relationship between ART duration and HIV viral load failures has not been consistent [33]. Studies in Gabon [34] and Mozambique [35] showed that clients on treatment for >5 years had increased risk of viral load failure. Therefore, further research is required to confirm these contradictions in the association between patients' number of years on antiretroviral therapy (ART duration) and virological failure.

The present study found that patients on the Efavirenz-based (TLE) combination were more likely to show viral load failure. This finding contradicts other studies, which highlighted the benefits to patients on TLE regimen in terms of ensuring adherence and achieving viral load suppression [32]. However, this study was done in 2016, before the WHO's approval to move from high Efavirenz-based to Dolutegravir-based regimen. To enhance viral load suppression in patients on antiretrovirals, health authorities should ensure the availability of the new regimen, educate clients on its efficacy, and provide sources for patients to report common side effects. This will ensure adherence and reduce viral load failure to achieve the new UNAIDS target of 95% by 2030.

Findings in this study identified the number of doses missed as a key predictor of virological failure in patients who missed two or more doses within a month. This result is in accordance with the literature in North Eastern Ethiopia [12] and South Eastern Ethiopia [36], where patients who missed their appointments were likely to miss their doses and increase the odds of virological failure. Other studies from Myanmar [37] and Australia [38] also agreed with this finding, that missed appointments caused missed doses among patients on ART. Doses missed might be caused by patients running out of medication due to the far distance to the ART site, which results in poor adherence to treatment and increases health issues for patients. To reverse this situation, health authorities and HIV support groups should enhance differentiated service delivery (DSD) and client-demanded delivery to ensure the regular and timely supply of ARVs to patients.

There was a high antiretroviral therapy adherence levels among respondents. This is similar to the findings of a study conducted in the Democratic Republic of Togo, which also found 62.6% ART adherence among patients [39]. However, the prevalence reported in the present study is lower than the 75.4% antiretroviral therapy adherence reported in Iran [40] and 66% ART adherence in Brazil [41]. Contrarily, antiretroviral therapy adherence among patients in South East Nigeria was reported to be (25%) [42], which is lower than the current finding. These discrepancies might be due to the study instruments, individual behavior, and culture in the various study settings.

As high as 97.3% of the respondents reported that they received adherence counseling. In contrast, a study in Ethiopia achieved 66.4% viral suppression after enhanced adherence counseling [43]. The differences could be attributed to the different approaches and policies in the treatment of newly diagnosed patients. The two goals of adherence counseling are to evaluate adherence once ART has begun and to assess the patient's preparedness for treatment before the commencement of the therapy [44]. To optimize patients' success in their treatments, adherence counseling should be enforced.

The viral suppression rate was slightly above 50% in patients on antiretroviral therapy. Although the current finding is lower compared to the set target of the UNAIDS 90% for 2020 in the Upper East Region, the proportion with virological failure (47.0%) among the patients represented a better performance compared to the Ghana AIDS Commission reported rate of 49% viral load failure in the region [45]. However, compared to other studies conducted across Ghana, the viral failure rate (47.0%) was the highest. For instance, in a study in Kumasi, Ashanti Region of Ghana, 23.6% viral load failure was reported [10]; in Ho, Volta Region of Ghana, a study found 31.3% viral load failure among HIV-positive clients on antiretroviral therapy [16]. The differences could be associated with how frequently the medication is supplied to the regions for distribution. In this study, the failure rate could be connected to the high rate of patients running out of medication. However, a review in Zimbabwe found 14% high viral load among clients on HIV treatment. The wide variation could be attributed to the differences in approach adopted by different countries aimed at combating HIV/AIDS infection as well as literacy levels. In addition, variations in HIV patients in achieving virological suppression based on availability and accessibility to antiretroviral therapy in different countries could also explain this discrepancy. Globally, most developing countries encounter difficulties in ensuring universal access to antiretroviral drugs for people living with HIV. Therefore, to ensure the realization of the UNAIDS set target of 95% viral suppression in HIV-burdened countries, it is prudent for WHO and

for UNAIDS to increase support for the highly burdened countries, including Ghana, with antiretroviral drugs to achieve universal coverage.

As with all studies, this study has its own strengths and weaknesses. The foremost strength in the research was the inclusion of all public hospitals in the Upper East Region, except Bawku Presbyterian Hospital due to the conflict in Bawku. The limitations of this study included the use of only one viral load test result per client collected in 2020. Inclusion of previous results for patients who completed two (2) or more VL tests could have increased the suppression rate. Secondly, with the use of consecutive sampling techniques, there could be selection bias such that more people responded positively as they were proud of their adherence and improving health. To eliminate or reduce the issues of selection bias, the data were collected over the period of five months to ensure that those who were given two to three months' time to return for review could be captured. In addition, in other to obtain accurate data from the patients, ART data managers and clinicians who already were familiar with these patients were used to collect the data to ensure confidentiality. People were encouraged to participate in the study without pressure. These limitations, however, did not have negative effects on the findings of this study. The findings emanating from this study are practical and essential to the achievement of the viral suppression among HIV/AIDS patients in Ghana and contribute to the world literature on HIV/AIDs management.

## 5. Conclusions

Though the majority of the respondents adhered to ART, the proportion of viral load achievement fell short of the UNAIDS 90% suppression target for 2020. Strategic health interventions could help attain the new UNAIDS target of 95% viral suppression by 2030 in the Upper East Region of Ghana. Educational status, salary/income, ART duration, ART combination regimen, and number of doses missed were statistically significantly associated with virological failure in patients on antiretroviral therapy in the Upper East Region. There is the need to strengthen the ongoing social behavior change communication among patients on ART to enhance adherence in order to attain the new UNAIDS target of 95% viral suppression by 2030 in the Upper East Region of Ghana. In addition, the National AIDS Control Programme (NACP) should ensure a regular supply of medication to the Upper East Region.

**Author Contributions:** Conceptualization: A.A., H.I. and M.A.O.M.; Methodology: A.A., H.I., M.A.O.M. and M.N.A., Validation: A.A. and M.A.O.M.; Formal analysis: M.N.A.; Investigation: A.A., M.A.O.M., H.I. and M.N.A., Resources: M.A.O.M.; Data curation: M.N.A. and M.A.O.M.; Writing—original draft preparation: M.N.A.; Writing—review and editing: A.A., H.I., M.A.O.M. and M.N.A.; Visualization: A.A. and M.N.A.; Supervision: A.A. and H.I.; Project administration: M.A.O.M. and M.N.A. All authors have read and agreed to the published version of the manuscript.

**Funding:** This research received no external funding.

**Institutional Review Board Statement:** The study was conducted in accordance with the Declaration of Helsinki and approved by the Ghana Health Service Ethics Review Committee (GHS-ERC:047/03/22).

**Informed Consent Statement:** Written informed consent was obtained from all subjects involved in the study.

**Data Availability Statement:** The data supporting this article are available in the repository of University of Development Studies and will be made available on request to the corresponding author when required.

**Acknowledgments:** The authors are grateful to the study participants for their time.

**Conflicts of Interest:** The authors declare no conflict of interest.

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
