# Peer review of "Determinants of Virological Failure in HIV Patients on Highly Active Antiretroviral Therapy (HAART): A Retrospective Cross-Sectional Study in the Upper East Region of Ghana"

_venereology, doi:10.3390/venereology2010002_

Round 1
Reviewer 1 Report
The following comments and questions which needs further clarification were given for the authors.

Author Response
Reviewer #1
Reviewer #1 comment: What is the difference between multiple logistic regression analysis and multivariable analysis? –
Authors response: The multiple logistics regression is same as multivariate logistic regression.
Reviewer #1 comment: Your abbreviation aOR not consistent with the main document
Author’s response: For consistency, aOR as captured in the abstract have been changed to AOR. The AOR is consistent throughout the study
Reviewer #1 comment: – The first sentence of the conclusion in the abstract part is not consistent with the objective of your study similar for the conclusion. See the detail in conclusion part.
Author’s response: the main objectives modified. The conclusion portion of the abstract is also modified to answer our main results. The repeated use of HAART has been reduced and.
Reviewer #1 Comments: Background: - In general the introduction part is well written but lacks some information: for example - The global and the country (Ghana) level data are good but not well searched
Author’s response: Overall, the introduction has been improved.
Reviewer #1 comment: African level data. - I missed the magnitude of the problem (not well stated in the background) as well not well written about the preference of virological when we compared with immunological and / or clinical treatment failure measurements. - Majority of the introduction part seems review document of HIV/AIDs
Author’s response: The Africa HIV cases are now provided to strengthen the work.
Reviewer #1 comment: Some abbreviations not expanded during first use. Example HIV line35, HIV/AIDs line 4 and …… - Line 43 grammar error - Line 43 issues like 95-95-95 and 90-90-90 line 57 their meaning not easy for readers –
Author’s response: All abbreviations have been explained upon first usage. Also, the terminologies such as the 95-95-95 and 90-90-90 have been explained to give meaning to those who may not be familiar with these terms
Reviewer #1 comment: Introduction part paragraph 2(lines46-55) as you stated sever determinants already identified in the previous studies. What is the significance of your research? Why not plan the intervention for mentioned problems? - Line 82/83 no reference for their review –
Authors response: Although, we alluded to the fact that some studies have been done elsewhere,
we also indicated that:
“…. currently, in developing countries including Ghana, inadequate evidence exists on the determinant of virological failure among HIV patients on HAART, identifying and managing these factors causing viral load failure in HIV patients can lead to increased success of HIV treatment for undetectable levels of viral load as well as ensuring an improved quality of life of HIV/AIDS patients in the Upper East Region. It is against this backdrop this study seeks to identify the determinants of virological failure in HIV patients on antiretroviral therapy (ART) in the Upper East Region of Ghana and produce findings for informed policy and decision making to attain the 95-95-95 by 2030….” This in our view is the problem and the significance of our study.
We do agree an interventional study is good, however, we would need to get basic understating of the issues in the Ghanaian context first before attempting interventions. generally, the introduction has been improved and answers all your concerns. Thank you.
Reviewer #1 comment Last paragraph of the background too long and no reference for some sentences.
Authors response: Thank you for the positive comments on the introduction. Overall, the introduction has been improved.
Methods and Materials
Reviewer #1 comments:- 547 health facilities line 103, Are they the whole country or the study area health facilities? - Abbreviations in the method (study area part) not expanded.
Authors response: The Upper East Region has 547 health facilities 20 hospitals (1 regional hospital at Bolgatanga, 6 district and municipal Government hospitals, 10 private hospitals and 3 CHAG (Christian Health Association of Ghana) hospitals), 67 health centers, 38 clinics, 419 CHPS compound and 3 private maternity homes. Also, CHAG has been expanded on its first usage.
Reviewer #1 comments: Why you exclude age less than 15? - You already excluded by the inclusion criteria, why you rewrite? Age less than 15 and patients who have only one viral load count.
Authors response: The study was opened to all persons who came to utilized services at the ART centers in the region and were on antiretroviral therapy for ≥6 months with at least one (1) readily available viral load result. At the end of the data collection period, the minimum age of the respondents recruited into the study was 15 years. No person below 15 years was drawing service at any ART center at the time of the study
Reviewer #1 comments: What percent you added to compensate non response rate for sample size calculation (line125)? What is your reference? What is the possible maximum percent?
Authors response: There is no scientific cut offs or bench mark values for non-response rate. The calculated sample size was 353. By definition, the sample size is the minimum number of participants to be recruited for a study to be considered representative. Also, the sample size influences two statistical properties: 1) the precision of our estimates and 2) the power of the study to draw conclusions. Given that our sample size was more than 353, the precision of our estimates was considered statistical sound. Thus, any increase in sample size more than the original 353 is said to increase the power of the study to draw conclusions. Besides, we did not record any non-response. Non response or dropout rate in longitudinal studies affect adversely statistical analysis if it is more than 30%.
Reviewer #1 comments: (Line 129) you applied convenient sampling techniques. Why you used non probability sampling techniques? Since HIV patients have well documented data record.
Authors response: Although there were documented records, it was ethical wrong for the hospital to release them to us. Also, the patient will not be comfortable with strange people contacting them on their status and seeking to interview them. The peculiarities and the beliefs surrounding HIV/AIDS restricted our ability to use a probability sample. Instead, the authors employed the consecutive sample to ensure that all persons who visited the ART facilities and agrees to participate in the study after they have met the inclusion were engaged in a private room to answer the questionnaire. This process was repeated in all the centers until the sample size was reached To further ensure that the recruitment process was robust and with minimal biases, the period for data collection was extended for five months to ensure that even those who visit the center every three months were captured.
Reviewer #1 comments: Clinical staffs were involved from each facility (Line 155) why? Those working in study facility participated in the study. How did you avoid the bias?
Authors response: The clinicians and ART data managers were selected because they have been working with these patients. Thus, since there are stigma issues regarding HIV/AIDS and people perceiving same as personal, the study participants would be more comfortable with the health care providers. The enumerators were trained and encouraged not to interfere with the patient response. We believed that bias was minimized as much as possible.
Reviewer #1 comments: Line 188/189 in most counties HIV patients may have appointment more than a month may be 2 or 3 months, if the patient have three months appointment and missed 5 or more doses how did you calculate adherence rate?
Authors response: Using the WHO ART adherence guideline, within a month, patients who missed ≤ 2 doses of antiretroviral therapy were said to have good adherence level (≥95%), patients who missed 3 - 5 doses have fair adherence level (≥85% to <95%), and patients with ≥6 doses missed have poor adherence level (˂85%)[23,24]. In Ghana, the practice is for HIV patients to come to the ART facilities for refill every month. The appointment is booked at most 3 days before the 30 pills are exhausted.
Reviewer #1 comments: (Line 191) you applied the adherence tool using this reference (reference 23) but it is about “infant and children’s” but your study population were adults age greater than 15. How? - Basic definition for viral load is three categories but the definition of not detected was different in the same document…….
Authors response: The references are updated and the right value is less than 50 copies. These have been corrected in the updated manuscript.
Reviewer #1 comments: Line 205, p-value less than 0.25 included for further analysis. Why you prefer 0.25? Why not other cut point or why not all?
Authors response: Chi-square is crucial in the goodness of fit of a logistic regression model. There is available literature suggesting that considering variables with p values less than 0.3 or p value closer to zero (0) gives a desirable outcome of the goodness of a logistic regression model(Archer et al., 2007). A p value of 0.3 is less than p value 0.25 as used in this manuscript. To control confounding variables and to achieved a goodness of fit for the model, only variables with a p-value of <0.25 were included in the multiple logistic regression. In another study by Abubakari et al., (2019), this approach was used with a p-value of 0.3 as benchmark variable. This cut off point is also essential in eliminating confounding variables.
Reviewer #1 comments: In the analysis part the model of analysis like AOR should be mentioned
Authors response: The output of bivariate and multiple logistic regression is COR and AOR respectively. Regardless, we have mentioned the AOR in the method sections.
Reviewer #1 comments: Lines 235-237 since they are observed in the table no need to write in the paragraph the whole results. - Lines in 257 and 259: opportunistic infection and drug resistance were 35.5% and 5.5% respectively for implication for failed viral load. If regimens are as such susceptible (few failure5.5%) do you expect as such large number (35.5%) of opportunistic infection occurrences? - Table 4 the result of P-value is that for AOR or COR?
Authors response:
Some narrations have been cancelled out.
In Lines 257 and 259, we did not have control of the response of the participants. This could rather form a good theme for future studies which the team is willing to chart that path in the near future.
Discussion
Reviewer #1 comments: Your discussion tries to address some points but you missed some points - As you mentioned in the first sentence the aim is “to determine the factors associated with virological failure in HIV patients on highly active antiretroviral therapy” but you missed to address your objectives. It seems the research which study about level of adherence. So, the first paragraph should answer the main finding of your study that is about virological failure and determinants. –
Authors response: The discussion has been modified
Reviewer #1 comments: Lines 310-317 not clearly written and no reference. It needs rewriting. - 322 how geographical location causes variation? - 331 no need to reason out for similar finding. - Line 374 it is better to compare by OR rather than P value. - Line 394 incomplete sentence –
Author’s response: Recommendation noted and affected portions revised accordingly.
Reviewer #1 comments: In general your searching strategy was shallow you missed some newly published HIV researches to compare the prevalence finding and determinants. Better to address researches like https://www.dovepress.com/virological-and-immunologicalantiretroviral-treatment-failure-and-pre-peer-reviewed-fulltext-article-HI
Authors response: On the literature search, we think our search have been exhaustive. We do agree also that we might have missed others which is normal. The suggested article was looked at and where we felt it was worthy of mention, we did same.
.
Conclusion and recommendation
Reviewer #1 comments:
- Your recommendation was not consistent with your finding. - You recommended what you did not address in your research objectives. Example: ✓ Behavioral change(abstract part) you did not studied behaviors ✓ Recommended to train qualified staff but not your finding ✓ About stoking issue,… about HIV testing and counseling….
Authors response:Recommendations have been modified
Reviewer 2 Report
This paper is basically a statistical analysis of data from Ghana. However, the statistics need to be revised significantly. The authors also include a number of variables as co-variates that overlap and confound one another. The subject the authors treat is a serious one that deserves a review and could impact public policy regarding HIV in Ghana. However, in its current condition, the work needs considerable revision in the statistical analysis, preferably by a statistician.
As well, the English usage is stilted and many words (e.g., predicators, instead of the correct predictors) are used incorrectly.
Author Response
Reviewer #2
Reviewer #2 Comments: This paper is basically a statistical analysis of data from Ghana. However, the statistics need to be revised significantly. The subject the authors treat is a serious one that deserves a review and could impact public policy regarding HIV in Ghana. However, in its current condition, the work needs considerable revision in the statistical analysis, preferably by a statistician.
Authors response: Thank you for your valuable revision. The dataset together with the current manuscript was submitted to two independent statisticians in the University for Development Studies, Tamale campus. They have declared our analysis as exhaustive and answers our research question. Thank you.
Reviewer #2 comment: The authors also include a number of variables as co-variates that overlap and confound one another.
Author’s response: This comment was rather very general however, where necessary, revision was done after engaging with our statistician
Reviewer #2 Comments: As well, the English usage is stilted and many words (e.g., predicators, instead of the correct predictors) are used incorrectly.
Authors response: The authors read through the work to make editing to the language as used in this manuscript. Also, the work was given to a colleague at the NHS, UK and the UDS librarian to offer language support.
Round 2
Reviewer 2 Report
This version is a significant improvement on the original. Here I have a few comments that are specific rather than global.
1. Your sample size calculation comes from the Survey Monkey Calculator. While I use Survey Monkey also, I am not aware of the calculator and I don't see in your text references to statistical power or effect size-- 2 of the 4 principal components of a survey size calculation. I would recommend using the statistical power functions of SPSS or the pwr module in R to recalculate your sample size, which will allow you to present the statistical power (the probability of avoiding a Type II error) and the effect size (the difference in results between groups that you are desirous of finding).
2. You accepted consecutive willing PLWH at the clinics until the sample size was reached. This strategy is certainly practical. However, you need to include then what kinds of selection bias this type of strategy might introduce. For example, you may have encountered more people who were having trouble with their meds than people who were doing well and were adherent. Or, you may have encountered and included more people who wanted to be included (and therefore said yes to the request) because they were proud of their adherence and improving health. Such a commentary would help strengthen confidence of the reader that the sample represents PLWH in the region.
3. I've never seen this Kobo-Collect software. You should have at least a reference to it.
4. In your discussion, you speak about people in their first year of ART who experience drug failure. Is it possible that some of these individuals in fact never achieved a viremia free status. Showing a graph of the evolution of the viremia in your patients would help with the argument that the proportion you state actually failed, that is, became non-viremic and then failed.
5. A quibble, your reference 43 to a Brazilian article. I'm in Brazil. Never heard of this article, but there are many, many articles I haven't read. I would suggest you have a look on PubMed for articles, particularly survey articles by Ricardo S. Diaz or Mauro Schechter. These authors are world-wide recognized authorities. An ethical warning here: Diaz is the director of my lab and I work a lot with both of them.
Let me finish with 1 general comment: why would someone reading this outside of Ghana find this article interesting? What is the generality that this case study speaks to that we could apply in other regions, like Latin America where I am? Some commentary on that in your discussion will make this more powerful and substantial.
Author Response
Reviewer Comment: This version is a significant improvement on the original. Here I have a few comments that are specific rather than global.
Authors comment: Thank you very much for your assessment.
Reviewer comment: Your sample size calculation comes from the Survey Monkey Calculator. While I use Survey Monkey also, I am not aware of the calculator and I don't see in your text references to statistical power or effect size-- 2 of the 4 principal components of a survey size calculation. I would recommend using the statistical power functions of SPSS or the pwr module in R to recalculate your sample size, which will allow you to present the statistical power (the probability of avoiding a Type II error) and the effect size (the difference in results between groups that you are desirous of finding).
Authors response: Our check on the four key components of survey size calculations shows:
- “Population size. The population size is the total number of people in the population (target audience) you are looking to survey.
- Confidence level.
- Confidence interval (Margin of error) ...
- Sample proportion.
In the current study all these four are stated in the sample size calculation.
As for the statistical power, most online sample size calculators are designs taking into consideration a power of 80.0% or above. Although not stated, the statistical power already is taken care of.
Significance level is correlated with power: increasing the significance level increases statistical power. Given that our p value was pegged less than 0.05, we expected an acceptable power. Thus, the probability that we reject the null hypothesis while it is true is high by our significance level set. Invariably the probability of rejecting the null hypothesis while it is false is also guaranteed which then becomes our power. That said, Significance is thus the probability of Type I error, whereas 1−power is the probability of Type II error. We believe the current sample size determination is exhaustive. Thank you.
Finally, Statistical power is positively correlated with the sample size, which means that given the level of the other factors viz. alpha and minimum detectable difference, a larger sample size gives greater power. Our current sample size us considered high and representative, our power therefore is great (Suresh & Chandrashekara,2012).
Kindly find below some authorities on above subject for your perusal:
- https://sphweb.bumc.bu.edu/otlt/mph-modules/bs/bs704_power/bs704_power_print.html
- https://www.jospt.org/doi/pdf/10.2519/jospt.2001.31.6.307
- Suresh, K. P., & Chandrashekara, S. (2012). Sample size estimation and power analysis for clinical research studies. Journal of human reproductive sciences, 5(1), 7.
To bring finality to the sample size determination, we have changed the survey monkey calculator to Taro Yamane’s formula(Yamane, 1967). Which actually gives us the same answer. Thank you
Reviewer comment: You accepted consecutive willing PLWH at the clinics until the sample size was reached. This strategy is certainly practical. However, you need to include then what kinds of selection bias this type of strategy might introduce. For example, you may have encountered more people who were having trouble with their meds than people who were doing well and were adherent. Or, you may have encountered and included more people who wanted to be included (and therefore said yes to the request) because they were proud of their adherence and improving health. Such a commentary would help strengthen confidence of the reader that the sample represents PLWH in the region.
Author response: We have improved the limitation section.
Reviewer comment: I've never seen this Kobo-Collect software. You should have at least a reference to it.
Author response: Thank you for the input. Instead, it is “Kobo Toolbox mobile App” Details of this can be find in this link (https://www.kobotoolbox.org/). This is a popular software used widely by scholar.
Reviewer Comment: In your discussion, you speak about people in their first year of ART who experience drug failure. Is it possible that some of these individuals in fact never achieved a viremia free status. Showing a graph of the evolution of the viremia in your patients would help with the argument that the proportion you state actually failed, that is, became non-viremic and then failed.
Author responses: This would be interesting we must admit. However, our inclusion was those with at least one viral load results. The data collected was specific to answering the research questions. Regardless, this would be interesting and we have decided conducting a study in this line with this by the end first quarter of 2023.
Reviewer Comment: A quibble, your reference 43 to a Brazilian article. I'm in Brazil. Never heard of this article, but there are many, many articles I haven't read. I would suggest you have a look on PubMed for articles, particularly survey articles by Ricardo S. Diaz or Mauro Schechter. These authors are world-wide recognized authorities. An ethical warning here: Diaz is the director of my lab and I work a lot with both of them.
Author responses: The reference 43 have been cited over 50 times. Just like we have missed out of some important references which you have suggested for us to read and cite, you may have missed out on this. Regardless, we have read some papers by the names suggested including the one on “Pre-Exposure Prophylaxis Failure With a Multiple Drug-Resistant HIV-1 Clade C Virus in Brazil.”. Those authors are great authorities in HIV research. We shall resort to them in our numerous HIV works in the pipeline.
Reviewer comment: Let me finish with 1 general comment: why would someone reading this outside of Ghana find this article interesting? What is the generality that this case study speaks to that we could apply in other regions, like Latin America where I am? Some commentary on that in your discussion will make this more powerful and substantial.
Authors response: An attempt has been made in the discussion to enhance the commentary.
Thank you.